# Evaluation of 3-Deoxy-D-Arabino-Heptulosonate 7-Phosphate Synthase (DAHPS) as a Vulnerable Target in *Mycobacterium tuberculosis*

Luiza Galina,[a,b] Fernanda S. M. Hopf,[a,c] Bruno Lopes Abbadi,[a] Nathalia D. de Moura Sperotto,[a] Alexia M. Czeczot,[a,b] Mario A. Duque-Villegas,[a] Marcia Alberton Perello,[a] Letícia Beatriz Matter,[a,c] Eduardo Vieira de Souza,[a] Tanya Parish,[d] Pablo Machado,[a,c] Luiz A. Basso,[a,b,c] Cristiano V. Bizarro[a,c]

[a]Centro de Pesquisas em Biologia Molecular e Funcional (CPBMF) and Instituto Nacional de Ciência e Tecnologia em Tuberculose (INCT-TB), Pontifícia Universidade Católica do Rio Grande do Sul (PUCRS), Porto Alegre, Rio Grande do Sul, Brazil

[b]Programa de Pós-Graduação em Medicina e Ciências da Saúde, Pontifícia Universidade Católica do Rio Grande do Sul, Porto Alegre, Rio Grande do Sul, Brazil

[c]Programa de Pós-Graduação em Biologia Celular e Molecular, Pontifícia Universidade Católica do Rio Grande do Sul, Porto Alegre, Rio Grande do Sul, Brazil

[d]Center for Global Infectious Disease Research, Seattle Children's Research Institute, Seattle, Washington, USA

**ABSTRACT** Tuberculosis (TB) remains one of the leading causes of death due to a single pathogen. The emergence and proliferation of multidrug-resistant (MDR-TB) and extensively drug-resistant strains (XDR-TB) represent compelling reasons to invest in the pursuit of new anti-TB agents. The shikimate pathway, responsible for chorismate biosynthesis, which is a precursor of important aromatic compounds, is required for *Mycobacterium tuberculosis* growth. The enzyme 3-deoxy-d-arabino-heptulosonate 7-phosphate synthase (*Mtb*DAHPS) catalyzes the first step in the shikimate pathway and it is an attractive target for anti-tubercular agents. Here, we used a CRISPRi system to evaluate the DAHPS as a vulnerable target in *M. tuberculosis*. The silencing of *aroG* significantly reduces the *M. tuberculosis* growth in both rich medium and, especially, in infected murine macrophages. The supplementation with amino acids was only able to partially rescue the growth of bacilli, whereas the Aro supplement (aromix) was enough to sustain the bacterial growth at lower rates. This study shows that *Mtb*DAHPS protein is vulnerable and, therefore, an attractive target to develop new anti-TB agents. In addition, the study contributes to a better understanding of the biosynthesis of aromatic compounds and the bacillus physiology.

**IMPORTANCE** Determining the vulnerability of a potential target allows us to assess whether its partial inhibition will impact bacterial growth. Here, we evaluated the vulnerability of the enzyme 3-deoxy-d-arabino-heptulosonate 7-phosphate synthase (DAHPS) from *M. tuberculosis* by silencing the DAHPS-coding *aroG* gene in different contexts. These results could lead to the development of novel and potent anti-tubercular agents in the near future.

**KEYWORDS** vulnerability, CRISPRi, aroG, shikimate pathway, macrophages

Tuberculosis (TB), a disease caused mainly by the bacillus *Mycobacterium tuberculosis*, is one of the leading causes of death worldwide. Until the coronavirus (COVID-19) pandemic, TB was the leading cause of death from a single infectious agent. Unfortunately, the COVID-19 pandemic has reversed years of progress in providing essential services against TB and reducing the disease burden (1). Although TB is a curable disease, the emergence of multidrug resistant and extensively drug-resistant strains of *M. tuberculosis* require an urgent development and implementation of new drugs for future control of TB (2). The shikimate pathway is essential for the biosynthesis of aromatic compounds in bacteria and other organisms and its absence in

Address correspondence to Cristiano V. Bizarro, cristiano.bizarro@pucrs.br.

The authors declare no conflict of interest.

mammals makes it an attractive target for the development of antimicrobial agents (3) (Fig. S1). This pathway has been shown to be essential for the viability of *M. tuberculosis* by the disruption of *aroK* gene, which encodes the shikimate kinase enzyme (4). The first enzyme of shikimate pathway is the 3-deoxy-d-arabino-heptulosonate 7-phosphate (DAHP) synthase (DAHPS). DAHPS catalyzes the first committed step in the shikimate pathway by a stereospecific condensation of phosphoenolpyruvate (PEP) and d-erythrose 4-phosphate (E4P), forming DAHP and inorganic phosphate, and is a major control point for shikimate pathway flux (5). The *aroG* gene encodes the DAHPS in *M. tuberculosis* (*Mtb*DAHPS) whereas in *M. smegmatis* this enzyme (*Msg*DAHPS) is encoded by the ortholog *MSMEG_4244*. According to a pairwise alignment (6), *M. tuberculosis* and *M. smegmatis* share 87.3% identity and 94.0% of similarity between the protein sequences, suggesting a high conservation between these organisms (Fig. S2). The *Mtb*DAHPS was shown to be negatively feedback regulated by metabolites whose biosynthesis relies on the shikimate pathway, Tryptophan (Trp), Phenylalanine (Phe), and Tyrosine (Tyr), by means of a sophisticated mechanism of allosteric regulation (7, 8). Additionally, the *Mtb*DAHPS forms an enzymatic complex with the chorismate mutase, which is located at the branch point leading to Phe and Tyr biosynthesis. The interaction between the *Mtb*DAHPS and the chorismate mutase results in a significant activation of the latter enzyme (9–11). These findings point toward *Mtb*DAHPS playing an important role in regulating the metabolism of aromatic compounds in *M. tuberculosis* and, as a consequence, being an attractive target for drug development. Target-based drug development approaches are primarily based on molecular target essentiality for pathogen survival during the infection in a human host. Another crucial feature of a target is its vulnerability, which is defined by the magnitude of inhibition required to lead the pathogen to death even when the target inhibition is incomplete (12–14). Additionally, the target vulnerability is correlated with the level of drug–target engagement required to generate a pharmacological response (15). Among the methods to validate vulnerable targets, the clustered regularly interspaced short palindromic repeats interference (CRISPRi) is an efficient tool for generation of gene specific knockdown strains (16), providing a required predictable and titratable reduction of gene expression (17).

Here, we present the results obtained using a CRISPRi system in different culture conditions to evaluate *Mtb*DAHPS as a vulnerable target for mycobacterial drug development.

## RESULTS

**Growth of *M. smegmatis* under *aroG* gene silencing by CRISPRi.** To evaluate the impact of *aro*G gene repression on cell growth in mycobacteria, we first performed CRISPRi experiments using the model organism *M. smegmatis*. We selected three sequences in the nontemplate (NT) strand and located in the first half of *aroG* coding sequence next to functional PAMs to be targeted by three different sgRNAs (Fig. 1A).

The silencing of the positive-control gene *mmpL*3 led to a strong growth perturbation starting after 9 h of bacterial culture (Fig. 2A). As expected, silencing using the sgRNA containing the scrambled sequence did not affect the growth curve (Fig. 2B). CRISPRi silencing of the *aro*G (*MSMEG_4244*) gene did not induce changes in the growth profile, (Fig. 2C to E). The absence of any growth perturbation caused by the silencing of the *MSMEG_4244* gene was also observed using the phenotyping methodology (see Materials and Methods for details). After 3 days of incubation, no difference between CFU in the presence or absence of ATc was observed for cultures that had the gene of interest silenced, only for the positive control, *mmpL*3 (Fig. 2A to E).

**M. *tuberculosis aroG* gene silencing by CRISPRi.** Next, we evaluated DAHPS vulnerability by silencing the *aroG* gene directly in the virulent *M. tuberculosis* H37Rv strain. Similar to the studies with *M. smegmatis*, we selected three sequences in the NT strand in the first half of *M. tuberculosis aroG* coding sequence as targets of sgRNAs (Fig. 1B). As expected, silencing of the *inhA* gene encoding the vulnerable target enoyl-ACP reductase (InhA) lead to a strong growth perturbation after 120 h of induction (Fig. 3A), while silencing with the scrambled sgRNA did not cause any growth defect (Fig. 3B). Interestingly, sgRNAs targeting three different sequences inside the *M. tuberculosis aroG* gene caused a

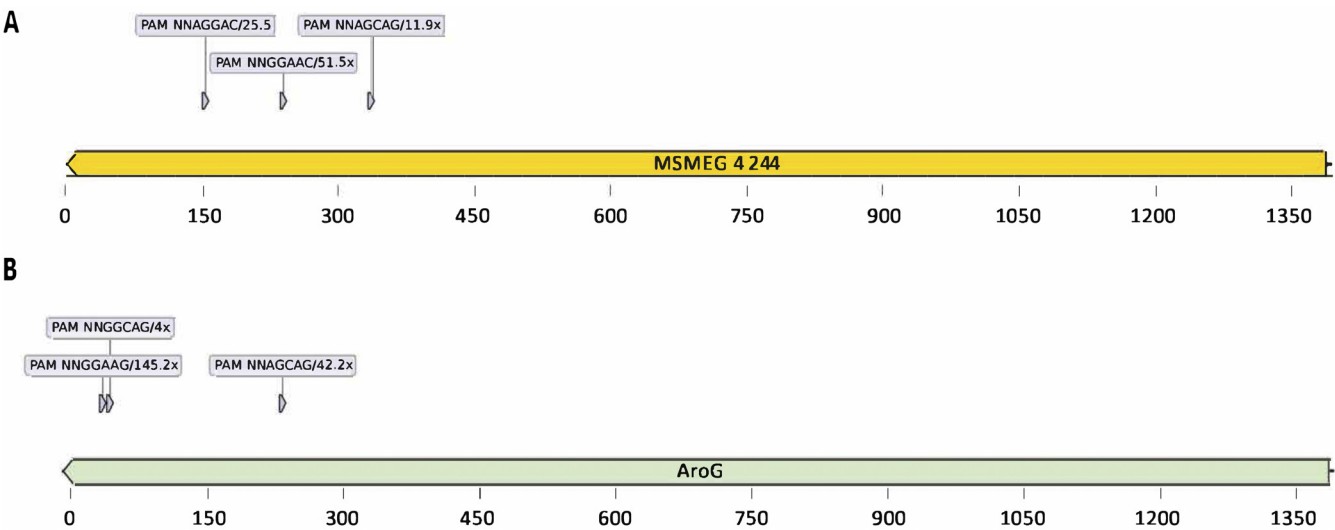

**FIG 1** Location of PAM sequences inside *MSMEG_4244* and *aroG* locus used in this study. (A) From left to right: 5′-NNAGGAC-3′, 5′-NNGGAAC-3′, 5′-NNAGCAG-3′. (B) From left to right: 5′-NNGGAAG-3′, 5′-NNAGCAG-3′, and 5′-NNGGCAG-3′. The repression strength of each PAM sequence, according to Rock et al. (2017) (16) is also described.

significant arrest in growth seen after 72 h for *Mtb*PAM1 and 96 h for *Mtb*PAM2 and *Mtb*PAM3 (Fig. 3C to E). This growth defect was sustained for at least 168 h (7 days). In accordance with the growth profile in liquid medium, phenotyping on solid medium also resulted in growth retardation in the *inhA*- and *aroG*-knockdown cells, but not in cells expressing the scrambled sgRNA (Fig. 3A to E).

We performed quantitative real-time (qRT-PCR) analysis to investigate the strength of the transcriptional repression on the *M. tuberculosis aroG* gene after 72 h of the CRISPRi system induction. The qRT-PCR analysis showed a strong transcriptional repression for all targeted sequences after adding 10 or 100 ng/mL of ATc (Fig. 4). Additionally, we did not observe any difference in *aroG* repression level regardless of the theoretical PAM repression strength.

We also performed CRISPRi-mediated *aroG* silencing experiments in media supplemented with aromatic amino acids ("aa" supplement; see Materials and Methods; Fig. 5A and C and E) or with the "aromix" supplement, containing the three amino acids and additional chorismate derivatives (see Materials and Methods for details; Fig. 5B, D, and F). Cultures grown in either solid or liquid media showed a partial restoration of growth in the presence of either supplement (Fig. 5 main and insets). Interestingly, bacterial growth was boosted more in the presence of the aa supplement than with the aromix supplement for all strains (Fig. S3). We determined the AUC obtained from all growth curves; in the presence of the aa supplement, the AUC increased from 48.86 to 65.49 for PAM1, from 56.28 to 69.26 for PAM2, and from 59.61 to 66.79 for PAM3. In the presence of the aromix supplementation, the AUC was decreased from 48.86 to 45.89 for PAM1, from 56.28 to 43.55 for PAM2, and from 59.61 to 49.91 for PAM3.

**Macrophage infection.** To investigate whether the *M. tuberculosis aroG*-targeted strains would survive in an environment that more closely resembles the host infection context, we performed CRISPRi silencing in an *in vitro* macrophage model of infection. Murine RAW 264.7 cells were infected with each *aroG*-targeted strain for a period of 3 h, at which point ATc was added to silence gene expression. As shown in Fig. 6, all three *aroG*-targeted strains showed a reduction in CFU ($P < 0.001$) after 3 days of ATc induction. Compared with the cultures without ATc, viability in the presence of ATc was reduced by 42.1% for PAM1, 28.3% for PAM2, and 39% for PAM3. These results suggest that *aroG* silencing jeopardizes *M. tuberculosis* growth in intracellular host conditions. Notably, the *aroG*-knockdown strains showed a similar intracellular growth perturbation pattern to the *inhA*-knockdown strain, which also was unable to sustain growth inside the macrophages (Fig. 6A and 6C–E). As expected, the knockdown strain containing the scrambled sgRNA maintained its viability in the macrophages during all the experiment (Fig. 6B).

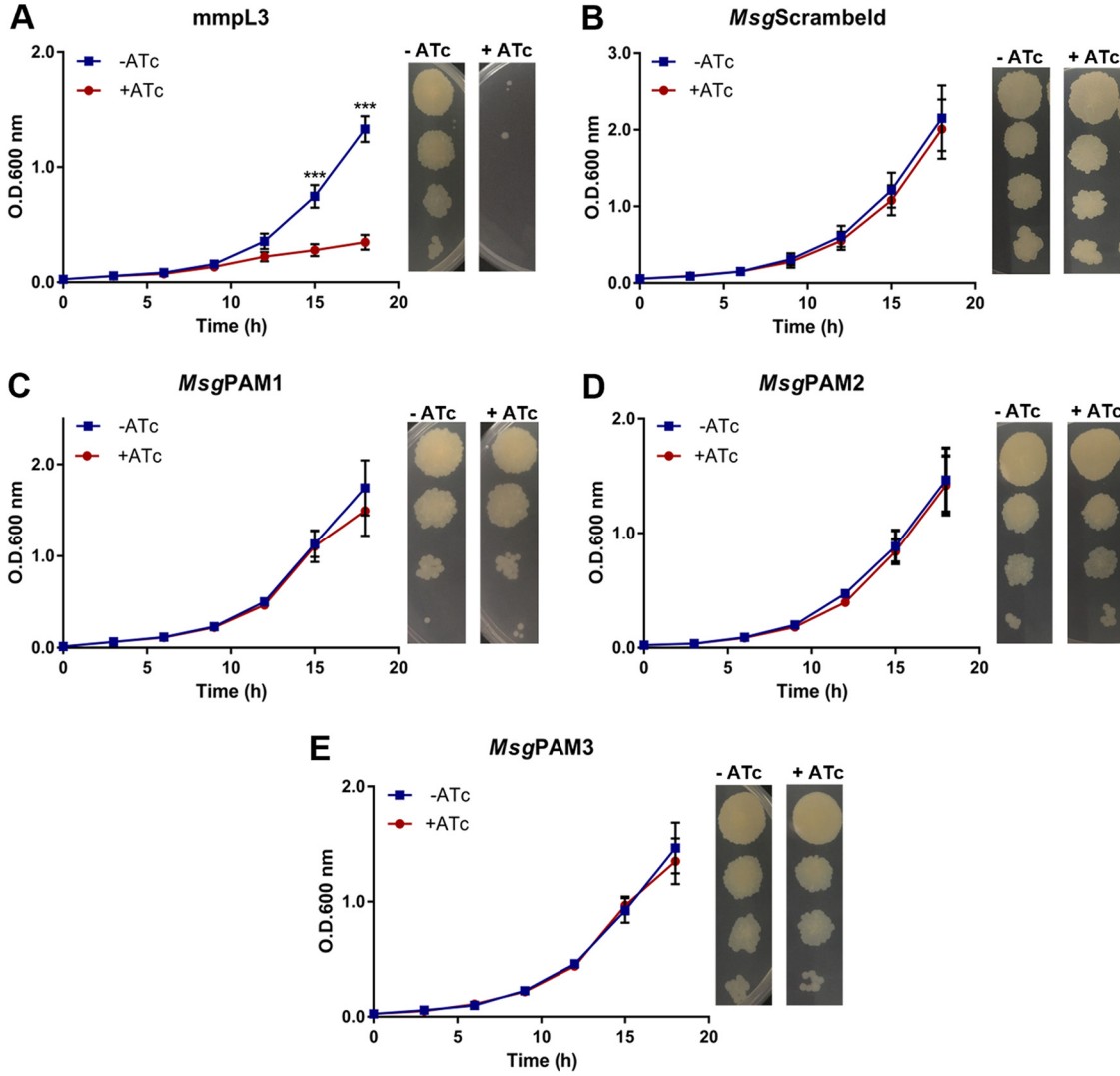

**FIG 2** *aroG* silencing in *M. smegmatis* does not affect growth *in vitro*. (A to E). *M. smegmatis* growth curves and dilution spots in the presence or absence of Atc 100 ng/mL. *mmpL3* (A) and scrambled sgRNAs were used as positive and negative controls, respectively (B). Three different *aroG* sequences were used *Msg*PAM1, *Msg*PAM2, and *Msg*PAM3 (C to E). Growth curves of *M. smegmatis* at 37°C in 7H9-OADC medium containing Atc 100 ng/mL. Cultures were inoculated to a starting $A_{600}$ of 0.07. Data are the mean $\pm$ standard deviation from biological and technical duplicates. Statistical analysis was performed using one-way ANOVA analysis followed by Bonferroni's posttest. ***, $P < 0.001$.

## DISCUSSION

In this work, we validated the vulnerability of the *aroG* gene under different experimental conditions and, more significantly, we proved that reduced expression of this gene affects the development of *M. tuberculosis* during macrophage infection. The gene vulnerability is defined as the level of gene repression needed to lead to a decrease in organismal fitness. Moreover, the target vulnerability is related to the levels of drug−target interaction to achieve the desired pharmacological outcome (14, 15). Studies of gene vulnerability have been used to infer the target vulnerability and select targets in drug discovery efforts. Recently, a CRISPRi-based system was developed to systematically titrate the expression of all annotated genes from *M. tuberculosis* and *M. smegmatis*, and to predict the vulnerability index (VI) of each gene. Those results are available on an online platform (https://pebble.rockefeller.edu) (17). The quantification of gene vulnerability expands our understanding upon the *M. tuberculosis* physiology and, therefore, distinguishes highly vulnerable genes from highly invulnerable genes for rational drug discovery (17). Nevertheless, it is necessary to evaluate gene vulnerability

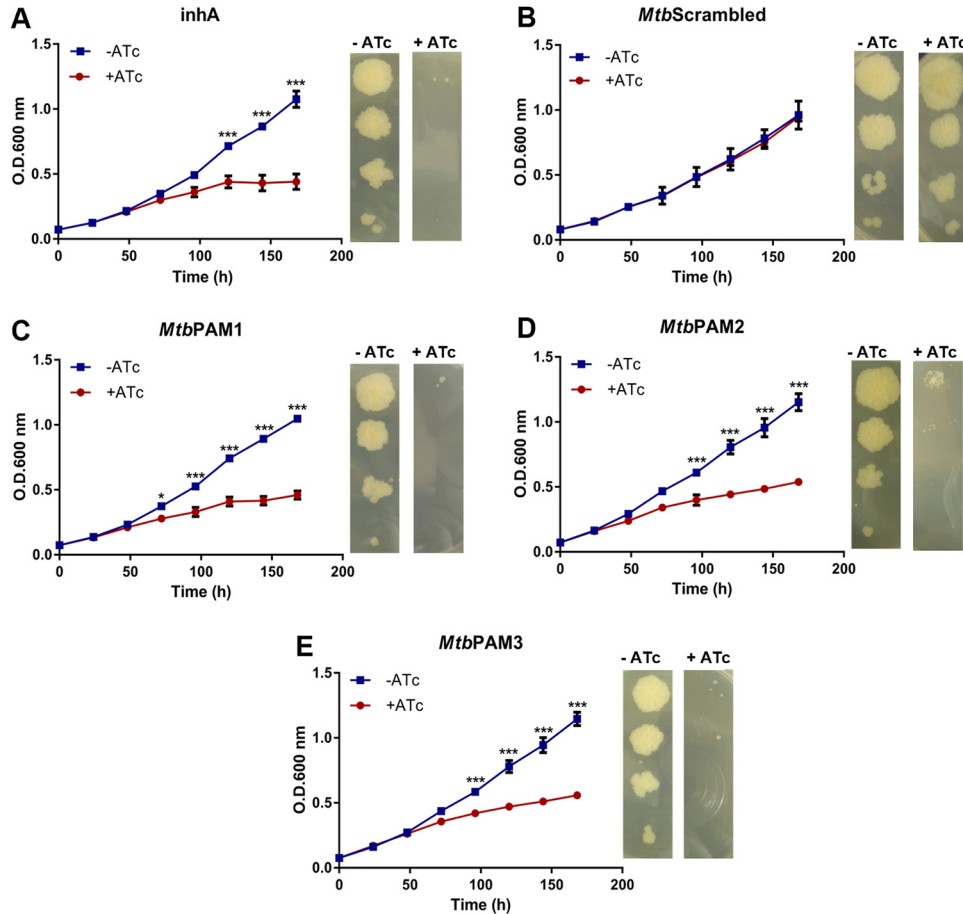

**FIG 3** *aroG* silencing in *M. tuberculosis* results in growth perturbation *in vitro*. (A to E). *M. tuberculosis* growth curves and dilution spots in the presence or absence of Atc 100 ng/mL. *inhA* (A) and scrambled sgRNAs were used as positive and negative control, respectively (B). (C to E) *aroG* was silenced using three different PAM sequences (*Mtb*PAM1, *Mtb*PAM2, and *Mtb*PAM3). Growth curves of *M. tuberculosis* at 37°C in 7H9-OADC medium containing Atc 100 ng/mL. Cultures were inoculated to a starting $A_{600}$ of 0.07. Data are the mean ± standard deviation from biological and technical duplicates. Statistical analysis was performed using one-way ANOVA analysis followed by Bonferroni's posttest. *, $P < 0.03$; **, $P < 0.002$; ***, $P < 0.001$.

under different growth environments and conditions. Also, it is important to validate whether the chemical inhibition of the target enzyme could lead to a corresponding impact on fitness, as observed by gene silencing (16, 18, 19). Gene knockdown approaches such as CRISPRi are worthy tools to investigate target vulnerability, and also provides an easy genetic strategy for the validation of novel drug targets and mechanisms of action that could facilitate the discovery of new anti-TB drug (16, 18, 19).

Due to its easier manipulation and faster grow, *M. smegmatis* has been used as an experimental model organism for mycobacteria, in particular for *M. tuberculosis*. At first glance, an evolutionary conserved pathway like the one leading to chorismate (Shikimate pathway), required for the essential activity of providing a precursor for the synthesis of basic cellular building blocks such as aromatic amino acids, seems to be well suited to be studied in this model system. However, a closer inspection on the data currently available for both gene essentiality and vulnerability in this pathway reveals a different picture (Table S2).

Aside *aroE* (Rv2552c), all the *aro* genes from *M. tuberculosis* were found to be both essential and vulnerable. In *Mycobacterium smegmatis*, on the other hand, the picture is completely different: no gene of the pathway was predicted to be either essential or nonessential, being classified as "uncertain" in terms of gene essentiality. In terms of gene vulnerability, the genes encoding enzymes of the last three steps of the pathway (*aroK*, *aroA*, and *aroF/aroC*) were vulnerable, while genes encoding enzymes catalyzing the first four steps were either invulnerable (*aroG/*MSMEG_4424, *aroQ/*MSMEG_2532, *aroK/*MSMEG_3031) or with intermediate

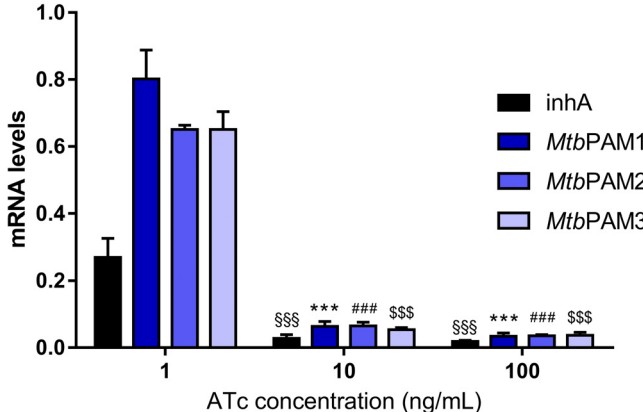

**FIG 4** mRNA levels in cultures expressing *inhA*, (control) or *aroG*-targeting sgRNAs induced with different levels of ATc. mRNA levels are expressed relative to a scrambled sgRNA control with 100 ng/mL ATc in *M. tuberculosis*. Results are the mean ± standard deviation of two experimental duplicates from two biological replicates. §§§, $P < 0.001$; ***, $P < 0.001$; ###, $P < 0.001$; $$$, $P < 0.001$.

VI values (*aroB*/MSMEG_3033, *aroD*/MSMEG_1922, *aroE*/MSMEG_3028). As can be seen from Table S2, two of the three invulnerable genes from *M. smegmatis* have a second related copy in the genome (MSMEG_1922 and MSMEG_2532 for *aroQ* and MSMEG_0453 and MSMEG_3031 for *aroK*). This redundancy could explain the lack of vulnerability for these two genes, but no additional copy or alternate enzyme is annotated for *aroG*, the focus of this study. As no other function aside producing chorismate is known for the shikimate pathway, this lack of consistency in terms of vulnerability for different genes of the pathway is hard to explain.

It is in this context that we have undertaken the task of revisiting the gene vulnerability of *aroG* in both bacterial species using a targeted, complementary approach to the one used for derivation of VI values. Our results showed that the *aroG* knockdown in *M. smegmatis* did not cause impairment on cellular growth (Fig. 2C to E). These results are congruent with the recent online vulnerability platform which revealed that *aroG* is, in fact, invulnerable in *M. smegmatis* (VI −0.2050) while *aroG* is a highly vulnerable gene in *M. tuberculosis*, with a VI of −13.3670 (17).

In our view, the data provided firmly determined that the differences in terms of *aroG* vulnerability between *M. smegmatis* and *M. tuberculosis* are real and should be studied further. It will be interesting to explore the *M. smegmatis* genome for the existence of alternate enzymes that could perform this activity. However, in the context of target validation for drug discovery purposes, these results served to redirect our efforts in experiments performed directly on *M. tuberculosis*.

Silencing of the *aroG* gene in *M. tuberculosis* impairs the growth of the bacillus to a significant extent (Fig. 3). According to the vulnerability predictions made by Bosch et al. (2021) (17), the *aroG* in *M. tuberculosis* shows a VI of −13.3670 whereas the *inhA*, which encode the target of the first-line drugs for TB, isoniazid, presents a VI of −9.9130. These differences in VI values could explain the earlier impact observed on our growth curves in *aroG* than in *inhA* in the presence of ATc (Fig. 3A and C). The effect of gene silencing on growth curves was confirmed at molecular level with the use of selective probes for qRT-PCR, as shown in Fig. 4. Regardless of the predicted silencing strength of the PAM, a strong effect on *aroG* expression levels was observed at 10 ng/mL of ATc. This correlated with the phenotypic effect where the same reduction in growth rate was seen between the strains.

Interestingly, when supplemented with the aromatic amino acids, the inhibition of bacterial growth was diminished (Fig. 3 and 5; Fig. S3). The amino acid intake can be partly explained by the fact that *M. tuberculosis* has an *aroP2* homologue, an aromatic amino acid transporter (20). A comparable result was observed by the silencing of the *aroA* gene in *M. smegmatis* (21). This gene encodes the enzyme 5-enolpyruvylshikimate-

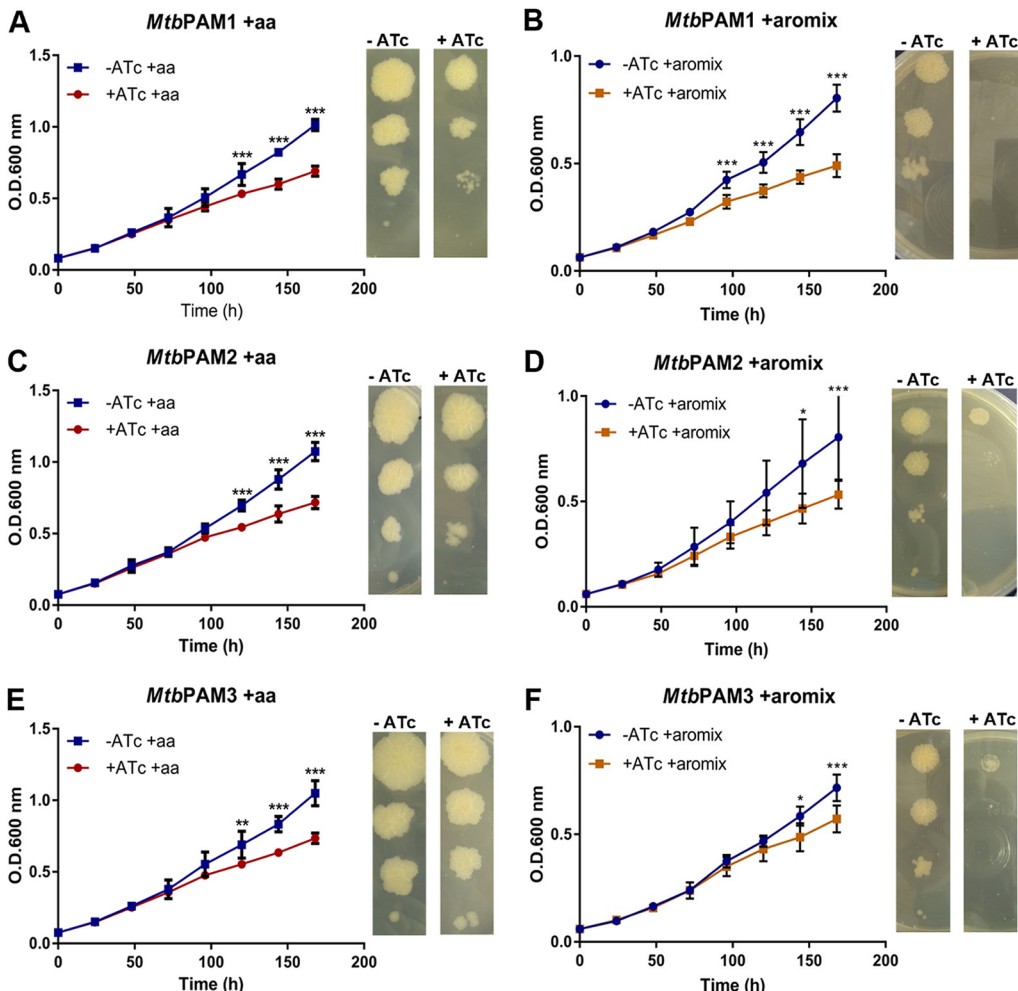

**FIG 5** Effect of *aroG* silencing on *M. tuberculosis* in supplemented media. (A, C, E). *M. tuberculosis* growth curves and dilution spots in the presence or absence of Atc 100 ng/mL with aa supplement. (B, D, F) *M. tuberculosis* growth curves and dilution spots in the presence or absence of Atc 100 ng/mL with the aromix supplement. Cultures were inoculated to a starting $A_{600}$ of 0.07. Data are the mean ± standard deviation from biological and technical duplicates. Statistical analysis was performed using two-way ANOVA analysis followed by Bonferroni's posttest. *, $P < 0.03$; **, $P < 0.002$; ***, $P < 0.001$.

3-phosphate synthase (EPSPS), the sixth enzyme of the shikimate pathway. The *aroA* silencing provoked an impairment of bacterial growth within 24 h, suggesting that the abrupt reduction in endogenous levels of the EPSPS enzyme does not cause bacterial death, but impairs growth. However, supplementation with only the three amino acids L-tryptophan, L-phenylalanine, and L-tyrosine was sufficient to rescue the growth in *aroA*-knockdown cells, implicating that *aroA* from *M. smegmatis* is essential only when sufficient amounts of L-tryptophan, L-phenylalanine, and L-tyrosine (AroAAs) are not available (21). On knockout studies with the shikimate kinase-encoding *aroK* gene from *M. tuberculosis*, it was demonstrated that the essentiality of the mycobacterial shikimate pathway cannot be circumvented by supplementation with aromatic compounds, such as amino acids, p-hydroxybenzoate, p-amino-benzoic acid, and 2,3-dihydroxybenzoate (4). Nonetheless, our findings showed the aromix supplement appears to slow the bacilli progress (Fig. 5; Fig. S3). This could be explained, hypothetically, owing to the adaptive stress generated by the low expression of the *aroG* gene that makes the strains hypersensitive to the extracellular supply of nutrients, reducing their growth rate even in rich media. Something similar was observed in histidine and tryptophan auxotrophs strains of *M. tuberculosis* that were unable to survive a single-amino-acid starvation but were able to endure long periods of complete starvation (22). For most organisms, chorismate is the

starting substrate for the p-aminobenzoate, p-hydroxybenzoate, and isochorismate, leading to folates, ubiquinones, naphtoquinones, menaquinones, and mycobactins, respectively, as well as for the pathways leading to phenylalanine, tyrosine, and tryptophan. However, *M. tuberculosis* is thought to be deficient in ubiquinone, using only menaquinone in the electron transport chain (23). Despite that, p-hydroxybenzoate (a precursor of ubiquinones) is also produced from chorismate in *M. tuberculosis*, in a single-step reaction catalyzed by chorismate pyruvate-lyase (24). In this organism, p-hydroxybenzoate seems to be used exclusively for the synthesis of p-hydroxybenzoic acid derivatives (p-HBADs), glycoconjugates that are components of the bacterial cell wall, and, in a limited number of strains, phenolic glycolipids (PGL) (24). It is feasible that *aroG*-knockdown strains are able to absorb those compounds from the medium or produce them through compensatory routes, in enough quantities to maintain cell growth. Those outcomes suggest that the *aroG* gene is vulnerable, but context and concentration-dependent, because supplying the products from the branches of the shikimate pathway was capable of partially restoring the bacterial growth.

Next, we explore the vulnerability profile of *aroG* during infection in murine macrophages. In infected macrophages and in the granuloma, *M. tuberculosis* encounters different types of stressors imposed by the host immune system, such as nutrient deprivation, low pH, and low oxygenation (12). Because the metabolism of the bacterium has to adapt to those changes depending on the context and stage of infection, it is essential to investigate whether the vulnerability of a target is maintained in different scenarios.

In murine macrophages, silencing of *aroG* reduced viable counts to a level similar to that of *inhA* silencing, indicating that decreased DAHPS expression has a strong effect on *M. tuberculosis* survival in macrophages (Fig. 6). These data also suggest that the bacteria cannot obtain aromatic substrates in sufficient quantities to sustain intracellular replication. Therefore, the strong intracellular growth inhibition displayed by the *aroG*-knockdown strains appears to be a direct consequence of the inhibition of the *Mtb*DAHPs enzyme expression.

*M. tuberculosis* survival inside macrophages relies on the ability of the bacteria to acquire nutrients from the host cell. A systems-based tool for exploring the nitrogen metabolism of intracellular bacteria revealed that *M. tuberculosis* cometabolizes multiple carbon sources during intracellular growth in the human host cell (25). The macrophages acquire the nitrogen sources Aspartate (Asp), Glutamate (Glu), Glutamine (Gln), Leucine (Leu), Alanine (Ala), and Glycine (Gly) directly from the growth media. *M. tuberculosis* takes Glu/Gln from the macrophage via an as yet unidentified transporter. Asp and asparagine is accessible to *M. tuberculosis* from the host macrophages via uptake by the AnsP1 transporter (26). Leucine/isoleucine (Leu/Ile) and Valine (Val) are acquired from the host macrophages via unidentified branched-chain amino acid, probably an ATP-binding cassette-type transporter. Ala, Gly, and Ser are possibly acquired via d-serine/d-alanine/glycine *transporter* system. Gln, Val, and Asp are potential nitrogen donors for cellular protein synthesis, with Gln as the principal nitrogen donor in intracellular *M. tuberculosis*. The nitrogen flux analysis study revealed that several amino acids, including Phe and Tyr, are synthesized *de novo* to suit biomass requirements for growth during macrophage infection (25). Analysis of the nonsteady state metabolism of carbon in intracellular *M. tuberculosis* demonstrated that the macrophages imported essential amino acids from the RPMI medium but made nonessential amino acids *de novo*, whereas the *M. tuberculosis* amino acids were all synthesized from host-derived substrates (either *de novo* or directly incorporated). The data also showed that the macrophage amino acids Ala, Glu/Gln, and Asp contribute to the intracellular nutrition of *M. tuberculosis* (27).

Another study highlighted that bacteria survive host CD4-generated stress through the production of Trp, thus avoiding starvation and death. It was demonstrated that loss or inhibition of the Trp biosynthetic pathway renders *M. tuberculosis* hypersusceptible to IFN-γ-mediated killing within macrophages, both *in vitro* and during infection (28). Consequently, our findings suggest that the *aroG* silencing can compromise fitness of the bacilli and the biosynthesis of chorismate derivatives

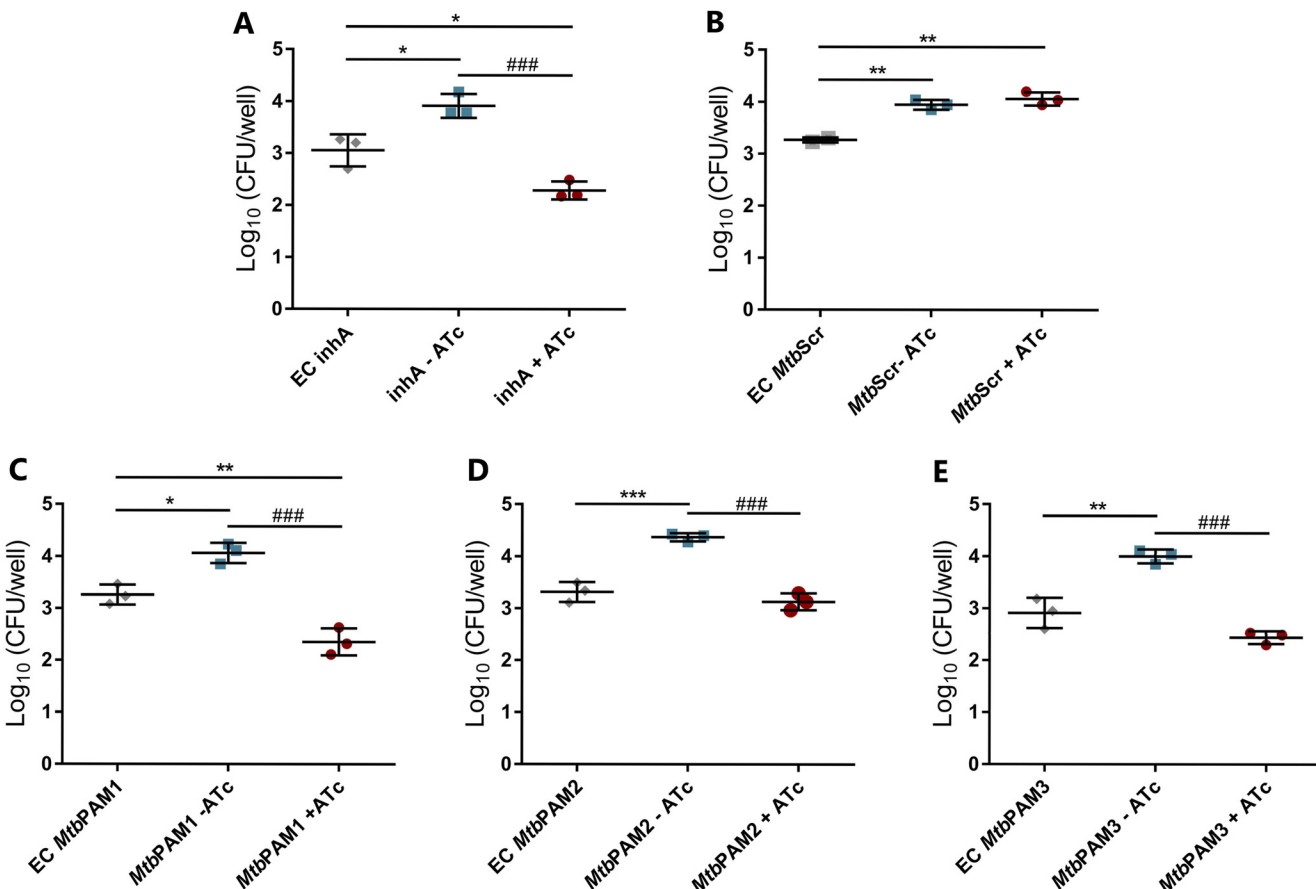

**FIG 6** Evaluation *M. tuberculosis aroG* silencing in macrophages. Comparative analysis of intracellular growth profile of controls (A, B) and *aroG* knockdown strains of *M. tuberculosis* (C, E). Intracellular bacterial load was determined by counting viable bacteria (CFU). EC, early control; *Mtb*Scr, scrambled. Error bars represent the SD from at least three measurements. Statistical significance is determined by using one-way ANOVA analysis followed by Bonferroni's posttest. *, $P < 0.03$; **, $P < 0.002$; ###, $P < 0.001$.

such as Phe, Tyr and, mainly, Trp, which could make the mycobacteria susceptible to CD4-mediated host defenses as well. According to our results, the *aroG* gene from *M. tuberculosis* remains vulnerable in this intracellular model. These data together reinforce the view of the *Mtb*DAHPS as a promising target for the development of antimicrobial agents.

In conclusion, this study contributes to the understanding of *aroG* function in *M. tuberculosis* and opens a possibility for rational drug design targeting the enzyme *Mtb*DAHPs. The inhibition of *aroG* expression was able to reduce *M. tuberculosis* in both rich media and, more importantly, during macrophage infection. This work also points to important differences in metabolic requirements for aromatic compounds between *M. smegmatis* and *M. tuberculosis*. This in turn highlights the drawbacks of using *M. smegmatis* as an experimental model in TB research, due to important physiologic and structural discrepancies between both species.

## MATERIALS AND METHODS

**Bacterial strains and culture conditions.** *Escherichia coli* DH10B strain was used for all cloning procedures and routinely grown in LB medium (broth and agar), at 37°C. All *M. tuberculosis* strains are derivatives of H37Rv (ATCC 27294) and all *M. smegmatis* strains are derivatives of mc²155. *M. tuberculosis* and *M. smegmatis* were routinely grown at 37°C in 7H9 supplemented medium (Difco Middlebrook 7H9 broth supplemented with 10% oleic acid-albumin-dextrose-catalase [OADC; BD, Difco], 0.2% glycerol [MERCK], 0.05% Tween 80 [Sigma-Aldrich], and 25 $\mu$g/mL kanamycin [Sigma-Aldrich], under constant shaking at 100 rpm for *M. tuberculosis* or 180 rpm for *M. smegmatis*. Solid cultures were grown in 7H10 agar supplemented medium (Difco 7H10 Agar [BD] supplemented with 10% OADC, 0.5% glycerol, and 25 $\mu$g/mL kanamycin at 37°C). For induction of the CRISPRi system, anhydrotetracycline (ATc, Sigma-Aldrich) dissolved in DMSO at 100 ng/mL

**TABLE 1** Oligonucleotides used for sgRNA CRISPRi system

| PAM sequence and fold repression | Primers forward | Primers reverse |
| --- | --- | --- |
| *Mtb*PAM1_NNGGAAG (145,2x) | 5′ gggagtgttcgcaggtcagtcggcagcg 3′ | 5′-aaaccgctgccgactgacctgcgaacac-3′ |
| *Mtb*PAM2_NNAGCAG (42,2x) | 5′ ggaaatgtctcagcgcagtcgccgccct 3′ | 5′-aaacagggcggcgactgcgctgagacatt-3′ |
| *Mtb*PAM3_NNGGCAG (4,0x) | 5′ gggagcgtccagtcgtgttcgcaggtcag 3′ | 5′-aaacctgacctgcgaacacgactggacgc-3′ |
| *Msg*PAM1_NNGGAAC (51,5x) | 5′ gggagcgcagtcaccgccctgcag 3′ | 5′-aaacctgcagggcggtgactgcgc-3′ |
| *Msg*PAM2_NNAGGAC (25,5x) | 5′ gggaggtgaccggcggcacgctct 3′ | 5′-aaacagagcgtgccgccggtcacc-3′ |
| *Msg*PAM3_NNAGCAC (11,9x) | 5′ gggaaccggcatgctcgcgccgtagg 3′ | 5′-aaaccctacggcgcgagcatgccggt-3′ |
| *inhA*_NNAGAAT (216.2x) | 5′ gggagtcggtgatgattccgctaa 3′ | 5′ aaacttagcggaatcatcaccgac 3′ |
| *mmpL3*_NNAGAAA (158.1x) | 5′ gggagcgacagatggctgccctcgtc 3′ | 5′ aaacgacgagggcagccagtctgtcgc 3′ |

was added. *E. coli* DH10B cells were transformed by electroporation (200 Ω resistance, 25 $\mu$F capacitance, and pulses of 2.5 kV) using 0.2 cm cuvettes. Mycobacterial cells were also transformed by electroporation following standard procedures (29), but the resistance was increased to 1,000 Ω. Cultures were allowed to recover in 10 mL 7H9 supplemented medium (see above) for approximately 4 h or 24 h with shaking at 37℃. Cultures were harvested, diluted, and plated onto 7H10 agar supplemented medium (see above). Plates were incubated at 37℃ for 4 days or 3 to 4 weeks.

**Construction of plasmids for the CRISPRi system.** The pLJR962 (*M. smegmatis*) and pLJR965 (*M. tuberculosis*) plasmids, carrying a CRISPR interference (CRISPRi) system developed by Rock et al. and reported in 2017 (16), were kindly provided by Sarah Fortune (Harvard University, USA).

The small guide RNAs (sgRNA) sequences were identified using an in-house script written in Python and made publicly available in the GitHub repository (https://github.com/Eduardo-vsouza/sgRNA_predictor). Three small-guide RNAs (named PAM1, PAM2, and PAM3) scaffolds were built for *M. smegmatis* and another three for *M. tuberculosis* to target different regions inside the coding sequence of the *aroG* gene (Rv2178c for *M. tuberculosis* and MSMEG_4244 for *M. smegmatis*), where three different PAM (protospacer adjacent motif) sequences were located (Fig. 1). PAM sequences selected for *M. smegmatis* were 5′-NNGGAAC-3′, 5′-NNAGGAC-3′, and 5′-NNAGCAG-3′, named as *Msg*PAM1, *Msg*PAM2 and *Msg*PAM3, having 51.5X, 25.5X and 11.9X fold repression, respectively, as predicted by Rock, et al. (16). For *M. tuberculosis*, the PAM sequences selected were 5′-NNGGAAG-3′, 5′-NNAGCAG-3′, and 5′-NNGGCAG-3′, named as *Mtb*PAM1, *Mtb*PAM2 and *Mtb*PAM3, having 145X-, 42.2X-, and 4.0X-fold repression, respectively. sgRNAs targeting the *mmpL3* (MSMEG_0250) or *inhA* (Rv1484) were used as positive controls of the gene silencing technique (see Table 1). A scrambled 22-bp sequence (5′-GGAGACGATTAATGCGTCTCGG-3′) not found in the genomes of neither *M. smegmatis* nor *M. tuberculosis* was included as a nontargeting negative control. To clone the variable sequence of the sgRNAs, a pair of complementary primers were designed (20 to 25 nt in length) to contain 5′ overhangs that are able to base pair with Esp3I restriction sites left in the Esp3I-digested plasmids. Recombinant plasmids were checked by DNA sequencing and restriction enzyme digestion. All sgRNA targets and PAM sequences used are listed in Table 1.

**Growth curves.** Growth curves were performed to monitor the effect of conditional silencing of the target genes on *M. tuberculosis* and *M. smegmatis*. For *M. tuberculosis*, a single colony of each strain carrying the CRISPRi system was grown in 10 mL of Middlebrook 7H9 broth at 37℃ to an optical density at 600 nm (OD$_{600}$) of 0.6 to 0.8. Bacterial cultures were then diluted to a theoretical OD$_{600}$ of 0.07 in fresh Middlebrook 7H9 containing kanamycin. The cultures were then equally divided (16 mL) in two conical tubes of 50 mL, with or without 100 ng/mL ATc before being incubated at 37℃ with shaking at 100 rpm. ATc was added to a final concentration of 100 ng/mL every 48 h and bacterial growth was monitored by measuring the OD$_{600}$ for 7 days.

For *M. smegmatis*, a single colony of each *MSMEG_4244*-targeted strain was grown in 5 mL of Middlebrook 7H9 broth containing kanamycin at 37℃ for 72 h. These cultures were then diluted (1:200) in fresh Middlebrook 7H9 containing kanamycin and grown overnight. The bacterial cultures were diluted to a theoretical OD$_{600}$ of 0.02 in fresh Middlebrook 7H9 and then equally divided (16 mL) in two conical tubes of 50 mL with or without ATc 100 ng/mL. Cultures were incubated at 37℃ with shaking at 180 rpm and bacterial growth was monitored by measuring the OD$_{600}$ every 3h for 18 h. The *mmpL3* (*M. smegmatis*) or *inhA* (*M. tuberculosis*) targeted strain was used as a positive control and the strain containing the sgRNA with the scrambled sequence was used as a negative control.

To evaluate the effect of supplementation with chorismate derivatives in the growth of *M. tuberculosis aroG*-targeted strains, growth curves were performed with 50 $\mu$g/mL of each of ʟ-tryptophan, ʟ-phenylalanine, and ʟ-tyrosine and 250 $\mu$M (each) 4-hydroxybenzoic acid, 4-aminobenzoic acid and 2,3-dihydroxybenzoic acid (aromix supplement) or with 50 $\mu$g/mL of each ʟ-tryptophan, ʟ-phenylalanine, and ʟ-tyrosine only (aa supplement).

These experiments were carried out using the same methodology described above. For cultures without ATc, 100 ng/mL of DMSO was added. Statistical analyses were performed using two-way ANOVA, followed by Bonferroni's posttest, using GraphPad Prism 9.0. Differences were considered significant at the 95% level of confidence. Results were expressed as mean ± standard deviation (SD) of two experimental duplicates from two biological replicates.

The area under the curve (AUC) was calculated using the R package Growthcurver, available for download at Comprehensive R Archive Network (CRAN) (30).

**Phenotyping.** A single colony of *M. smegmatis* or *M. tuberculosis* strain expressing each *aroG*-targeting sgRNA was grown to log-phase (OD$_{600}$ 0.6 to 0.8) in 7H9 broth. Then, cultures were diluted in fresh 7H9

broth to a bacterial density of 1,000, 100, 10, and 1 CFU/$\mu$L and deposited in isolated 5 $\mu$L spots (in triplicates) on 7H10 media with or without ATc 100 ng/mL. Plates were incubated at 37°C for 4 days (*M. smegmatis*) or 21 days (*M. tuberculosis*). Phenotyping experiments with supplementation with aa supplement or aromix supplement was performed using the same methodology.

**Macrophage infection.** Macrophage infection experiments were performed based on previously described protocols (31, 32), with some modifications. The macrophage murine cell line RAW 264.7 (obtained from Banco de Células do Rio de Janeiro, Rio de Janeiro, Brazil) was cultured in RPMI 1640 medium (Gibco, Thermo Fisher Scientific) supplemented with 10% heat-inactivated fetal bovine serum (FBS) (Gibco) at 37°C with 5% $CO_2$. Macrophages were seeded in 24-well culture plates (TPP, Trasadingen, Switzerland) at a density of 5 $\times$ $10^4$ cells/well in RPMI 1640 medium (supplemented with 10% FBS) and incubated for 24 h at 37°C with 5% $CO_2$. The cells were then washed three times with RPMI 1640 to remove nonadherent cells. Infection of RAW 264.7 cells with CRISPRi *M. tuberculosis* was performed at 1:1 MOI (macrophage:bacteria) for 3 h at 37°C with 5% $CO_2$. Afterwards, infected cells were washed twice with prewarm 1X PBS to remove noninternalized mycobacteria. Cell of the early control (EC) were lysed on the day of infection onset with 1 mL of SDS 0.025% (dissolved in sterile 0.9% saline). Lysates were serially diluted and plated in 7H10 containing kanamycin 25 $\mu$g/mL for CFU enumeration after 3 to 4 weeks of incubation at 37°C. Subsequently, the infected cells were maintained in RPMI-FBS medium containing 100 ng/mL ATc for 3 days. Next, wells were washed with prewarm 1X PBS and cells were disrupted with SDS 0.025% to release intracellular bacteria. The supernatant of each well was serially diluted in 1X PBS and spread on 7H10 plates for CFU enumeration after 3 to 4 weeks of incubation at 37°C. For this experiment, the *inhA*-targeted strain was used as a negative growth control and the strain with the sgRNA containing the scrambled sequence as a positive growth control.

The results were expressed as mean numbers of the logarithms ($\log_{10}$) of CFU per well, and were evaluated with the one-way ANOVA analysis, followed by Bonferroni's posttest, using GraphPad Prism 9.0. Differences were considered significant at the 95% level of confidence.

**mRNA quantification.** Cultures of *M. smegmatis* and *M. tuberculosis* were grown to log phase and then diluted to theoretical $OD_{600}$ value of 0.1. Then, ATc was added at 1, 10, or 100 ng/mL ATc to induce the CRISPRi system. Cultures were grown at 37°C under stirring for 9 h (*M. smegmatis*) or 3 days (*M. tuberculosis*). Cells from each culture with equivalents $OD_{600}$ values were harvested by centrifugation, suspended in 1 mL of RNA Protect (Qiagen) and kept at $-80$°C (freezer). The same procedure was performed with the control strain (scrambled_sgRNA) of each mycobacterial species, which were induced with 100 ng/mL of ATc. For the RNA extraction, samples were centrifuged, the bacterial pellet was suspended in RLT Buffer (RNeasy, Qiagen), and cells were disrupted by bead beating (Lysing Matrix B, MP Biomedicals) with three pulses of 1 min and 2-min intervals on ice. The next steps for RNA extraction were performed according to the RNeasy kit extraction protocol. Genomic DNA residual contamination was digested with Turbo DNA-*free* kit (Invitrogen, Thermo Fisher Scientific) and cDNA was prepared with random hexamers following Super Script III First Strand kit instructions (Invitrogen, Life Technologies). Gene expression was evaluated by quantitative real-time PCR (qRT-PCR) specific PrimeTime qPCR Probes (Table S1) from IDT (Integrated DNA Technologies) and 7500 cycler (Applied Biosystems). Primers and probes were designed by IDT software (https://www.idtdna.com/pages/tools). Signals were normalized with housekeeping *sigA* and quantified using the Livak and Schmittgen (2-$\Delta\Delta$Ct) method (33). The statistical analysis was performed using a two-way ANOVA analysis, followed by Turkey's posttest, using GraphPad Prism 9.0. Error bars are 95% of confidence intervals of two experimental duplicates from two biological replicates.

## SUPPLEMENTAL MATERIAL

Supplemental material is available online only.
**SUPPLEMENTAL FILE 1**, PDF file, 0.8 MB.

## ACKNOWLEDGMENTS

We thank Sara Fortune and Jeremy Rock for providing the pLJR962 and pLJR965 plasmids. C.V.B., P.M., and L.A.B. would like to acknowledge financial support given by CNPq/FAPERGS/CAPES/BNDES to the National Institute of Science and Technology on Tuberculosis (INCT-TB), Brazil (grant numbers: 421703-2017-2/17-1265-8/14.2.0914.1). C.V.B. (310344/2016-6), P.M. (305203/2018-5), and L.A.B. (520182/99-5) are research career awardees of the National Council for Scientific and Technological Development of Brazil (CNPq). This study was financed in part by the Coordenação de Aperfeiçoamento de Pessoal de Nível Superior—Brazil (CAPES)—Finance Code 001.

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
