## [Reviewer comments · Microbiology Spectrum]

Microbiology Spectrum

Evaluation of 3-deoxy-D-arabino-heptulosonate-7-phosphate synthase (DAHPS) as a vulnerable target in *Mycobacterium tuberculosis*

Luiza Galina, Fernanda Hopf, Bruno Abbadi, Nathalia Sperotto, Alexia Czczot, Mario Duque-Villegas, Márcia Perello, Letícia Matter, Eduardo de Souza, Tanya Parish, Pablo Machado, Luiz Basso, and Cristiano Bizarro

Corresponding Author(s): Cristiano Bizarro, Pontifícia Universidade Católica do Rio Grande do Sul

Review Timeline:

Submission Date:	March 3, 2022
Editorial Decision:	April 4, 2022
Revision Received:	June 6, 2022
Accepted:	June 24, 2022

Editor: Silvia Cardona

Reviewer(s): The reviewers have opted to remain anonymous.

Transaction Report:

DOI: <https://doi.org/10.1128/spectrum.00728-22>

April 4, 2022

Prof. Cristiano Valim Bizarro
Pontifícia Universidade Católica do Rio Grande do Sul
Centro de Pesquisas em Biologia Molecular e Funcional (CPBMF) and Instituto Nacional de Ciência e Tecnologia em
Tuberculose (INCT-TB)
Porto Alegre, Rio Grande do Sul 90616-900
Brazil

Re: Spectrum00728-22 (Evaluation of 3-deoxy-D-arabino-heptulosonate-7-phosphate synthase (DAHPS) as a vulnerable target in *Mycobacterium tuberculosis*)

Dear Prof. Cristiano Valim Bizarro:

In particular, the reviewers indicated that the work is mostly confirmatory of previous findings. Therefore, I encourage you to emphasize the new knowledge in the revised version.

Link Not Available

Sincerely,

Silvia Cardona

Journals Department
Reviewer comments:

Reviewer #1 (Comments for the Author):

Evaluation of 3-deoxy-D-arabino-heptulosonate-7-phosphate synthase (DAHPS) as a vulnerable target in *Mycobacterium tuberculosis*

The authors target the Mycobacterial *aroG*, using CRISPRi. This enzyme catalyses the first dedicated step in the shikimate pathway which leads to the production of chorismate, a precursor of aromatic amino acids. The vulnerability of *aroG* in *M.*

tuberculosis has been reported before in a genome wide CRISPRi screen (Bosch and De Jesus et al., 2021) hence this study represents an experimental validation of those findings. Overall the science is sound and the data support the conclusions drawn.

Minor corrections

Ln 30: change to - aroG significantly reduces...

Ln 125: change to - Dr. Sarah...

Ln 303: change to - additional

Ln 342: change to - ... expands our understanding of M. tuberculosis physiology and...

Ln 462: change to - ... study highlighted the mechanisms...

Ln 468: change to - ... chorismate derivatives such...

Figure 3: symbols have not been explained in legend.

Ln 680: change to - and negative control respectively...

Major corrections/suggestions

The invulnerability of MSMEG_4244 when targeted is expected since it is predicted to be non-essential by Tn-seq and the vulnerability index supports this. For this reason I think this data should be in the Supplementary rather than as a main figure. Ln 352-365: This information is not necessary in the discussion since this is well known and its removal makes the discussion more concise.

It would be useful to include a schematic of the shikimate pathway as a supplementary figure.

Figure 6 is a duplication of data (represented in a different way) and if considered necessary by the authors should be moved to supplementary.

Reviewer #2 (Comments for the Author):

The aroG gene encodes DAHP synthase of M. tuberculosis, the first step of the shikimate pathway which is essential for biosynthesis of aromatic amino acids, folates, mycobactins and menaquinol in mycobacteria. The shikimate pathway is known to be essential and as a result, DAHP synthase as well as other enzymes of the pathway have been investigated as drug targets. This work verifies that downregulation of aroG mRNA levels is inhibitory to M. tuberculosis in vitro as well as during growth in macrophages. Growth inhibition can be partially overcome by addition of aromatic amino acids with further addition of folate and mycobactin biosynthetic intermediates not further restoring growth but possibly even resulting in some growth inhibition for reasons that are not explored. The finding that aroG knockdown is inhibitory is not novel. Rescue of growth with aromatic amino acids is not surprising. Nevertheless, the work is well done.

Minor comments

Fig. 3: include length of exposure (hours) in the legend (not mentioned in results section either)

Figure 4: inspection of the graphs does not really suggest that PAM1 is very different from PAM2 in terms of timing in inducing growth arrest.

The aromix supplement supplement consists of L-Trp, L-phe and L-tyr along with 4-hydroxybenzoic acid, 4-aminobenzoic acid and 2,3-dihydroxybenzoic acid. 4-aminobenzoic acid would be an intermediate in folate biosynthesis. 2,3-dihydroxybenzoic acid would be an intermediate in mycobactin biosynthesis but this would only be important under iron limitation. The 4-hydroxybenzoic acid is less clear since it's not an intermediate of menaquinol biosynthesis. Do we know that these do not inhibit an enzyme at 250uM? The concentration seems a bit high (combined it would be 750uM of benzoic acid derivatives added) although admittedly, rescue for Mtb is often only possible at high concentrations.

Staff Comments:

Preparing Revision Guidelines

- Point-by-point responses to the issues raised by the reviewers in a file named "Response to Reviewers," NOT IN YOUR COVER LETTER.
- Upload a compare copy of the manuscript (without figures) as a "Marked-Up Manuscript" file.
- Each figure must be uploaded as a separate file, and any multipanel figures must be assembled into one file.

- Manuscript: A .DOC version of the revised manuscript
- Figures: Editable, high-resolution, individual figure files are required at revision, TIFF or EPS files are preferred

Please return the manuscript within 60 days; if you cannot complete the modification within this time period, please contact me. If you do not wish to modify the manuscript and prefer to submit it to another journal, please notify me of your decision immediately so that the manuscript may be formally withdrawn from consideration by Microbiology Spectrum.

Evaluation of 3-deoxy-D-arabino-heptulosonate-7-phosphate synthase (DAHPS) as a vulnerable target in *Mycobacterium tuberculosis*

The authors target the Mycobacterial *aroG*, using CRISPRi. This enzyme catalyses the first dedicated step in the shikimate pathway which leads to the production of chorismate, a precursor of aromatic amino acids. The vulnerability of *aroG* in *M. tuberculosis* has been reported before in a genome wide CRISPRi screen (Bosch and De Jesus et al., 2021) hence this study represents an experimental validation of those findings. Overall the science is sound and the data support the conclusions drawn.

Minor corrections

Ln 30: change to – *aroG* significantly reduces...

Ln 125: change to – Dr. Sarah...

Ln 303: change to – additional

Ln 342: change to - ... expands our understanding of *M. tuberculosis* physiology and...

Ln 462: change to - ... study highlighted the mechanisms...

Ln 468: change to - ... chorismate derivatives such...

Figure 3: symbols have not been explained in legend.

Ln 680: change to – and negative control respectively...

Major corrections/suggestions

The invulnerability of MSMEG_4244 when targeted is expected since it is predicted to be non-essential by Tn-seq and the vulnerability index supports this. For this reason I think this data should be in the Supplementary rather than as a main figure.

Ln 352-365: This information is not necessary in the discussion since this is well known and its removal makes the discussion more concise.

It would be useful to include a schematic of the shikimate pathway as a supplementary figure.

Figure 6 is a duplication of data (represented in a different way) and if considered necessary by the authors should be moved to supplementary.

Evaluation of 3-deoxy-D-arabino-heptulosonate-7-phosphate synthase (DAHPS) as a vulnerable target in *Mycobacterium tuberculosis*

Response to Reviewers:

Reviewer #1 (Comments for the Author):

The authors target the Mycobacterial *aroG*, using CRISPRi. This enzyme catalyses the first dedicated step in the shikimate pathway which leads to the production of chorismate, a precursor of aromatic amino acids. The vulnerability of *aroG* in *M. tuberculosis* has been reported before in a genome wide CRISPRi screen (Bosch and De Jesus et al., 2021) hence this study represents an experimental validation of those findings. Overall the science is sound and the data support the conclusions drawn.

1 - Ln 30: change to - *aroG* significantly reduces...

Thank you for the suggestion, the text was modified accordingly.

2 - Ln 125: change to - Dr. Sarah...

The text was modified accordingly.

3 - Ln 303: change to – additional

Changed, as indicated.

4 - Ln 342: change to - ... expands our understanding of *M. tuberculosis* physiology and...

Changed, as indicated.

5 - Ln 462: change to - ... study highlighted the mechanisms...

Thank you for the suggestion, the text was modified accordingly.

6 - Ln 468: change to - ... chorismate derivatives such...

Thank you for the suggestion, the text was modified accordingly.

7 - Figure 3: symbols have not been explained in legend.

Thank you for the suggestion, the meaning of symbols was added in legend.

8 - Ln 680: change to - and negative control respectively...

Thank you for the suggestion, the text was modified accordingly.

9 - The invulnerability of MSMEG_4244 when targeted is expected since it is predicted to be non-essential by Tn-seq and the vulnerability index supports this. For this reason I think this data should be in the Supplementary rather than as a main figure.

We agree with Reviewer #1 that, when considered in isolation and based on data from a genome-wide approach published previously (PMID: 34297925), our vulnerability results for MSMEG_4244 were expected and rather confirmatory in nature. However, when considered in the context of our work, we think it would be better maintaining them in the main text of the manuscript.

Due to its easier manipulation and faster grow, *M. smegmatis* has been used as an experimental model organism for mycobacteria, in particular for *M. tuberculosis*. At a first glance, an evolutionary conserved pathway like the one leading to chorismate (Shikimate pathway), required for the essential activity of providing a precursor for the synthesis of basic cellular building blocks such as aromatic amino acids, seems to be well suited to be studied in this model system. However, a closer inspection on the data current available for both gene essentiality and vulnerability in this pathway reveals a different picture (Table R1).

Table R1. Gene essentiality and vulnerability for genes encoding enzymes from the Shikimate Pathway

M.tuberculosis				M. smegmatis			
gene	Rv number	Essentiality ¹	VI ²	gene	MSMEG number	Essentiality ³	VI ²
aroG	Rv2178c	essential	-13.36	aroG	MSMEG_4244	uncertain	-0.2
aroB	Rv2538c	essential	-9.2	aroB	MSMEG_3033	uncertain	-4.93
aroD	Rv2537c	essential	-5.92	aroQ	MSMEG_1922	uncertain	-3.52
				aroQ	MSMEG_2532	uncertain	0.11
aroE	Rv2552c	NonEssential	-9.01	aroE	MSMEG_3028	uncertain	-4.02
aroK	Rv2539c	Essential	-10.16	aroK	MSMEG_0453	uncertain	0.24
				aroK	MSMEG_3031	uncertain	-7.18
aroA	Rv3227	Essential	-9.00	aroA	MSMEG_1890	uncertain	-7.56
aroF/aroC	Rv2540c	Essential	-11.38	aroC	MSMEG_3030	uncertain	-9.05

¹TnSeq essentiality predictions for *M. tuberculosis* taken from DeJesus et al. 2017 (PMID: 28096490).

²VI: Vulnerability Index. Vulnerability predictions taken from Bosch & DeJesus et al. Cell. 2021 (PMID: 34297925).

³TnSeq essentiality predictions for *M. smegmatis* taken from Dragset et al. 2019 (PMID: 31388080).

Data obtained from <https://pebble.rockefeller.edu>.

Aside *aroE* (Rv2552c), all the *aro* genes from *M. tuberculosis* were found to be both essential and vulnerable. In *M. smegmatis*, on the other hand, the picture is completely different: no gene of the pathway was predicted to be either essential or nonessential, being classified as “uncertain” in terms of gene essentiality. In terms of gene vulnerability, the genes encoding enzymes of the

last three steps of the pathway (*aroK*, *aroA* and *aroF/aroC*) were vulnerable, while genes encoding enzymes catalyzing the first four steps were either invulnerable (*aroG/MSMEG_4424*, *aroQ/MSMEG_2532*, *aroK/MSMEG_3031*) or with intermediate VI values (*aroB/MSMEG_3033*, *aroD/MSMEG_1922*, *aroE/MSMEG_3028*). As can be seen from Table R1, two of the three invulnerable genes from *M. smegmatis* have a second related copy in the genome (MSMEG_1922 and MSMEG_2532 for *aroQ* and MSMEG_0453 and MSMEG_3031 for *aroK*). This redundancy could explain the lack of vulnerability for these two genes but no additional copy or alternate enzyme is annotated for *aroG*, the focus of this study. As no other function aside producing chorismate is known for the Shikimate Pathway, this lack of consistency in terms of vulnerability for different genes of the pathway is hard to explain.

It is in this context that we have undertaken the task of revisiting the gene vulnerability of *aroG* in both bacterial species using a targeted, complementary approach to the one used for derivation of VI values. As detailed in the main text of the manuscript, our data are in complete agreement with data previously published. In our view, the data provided put in firm grounds that the differences in terms of *aroG* vulnerability between *M. smegmatis* and *M. tuberculosis* are real and should be studied further. It will be interesting to explore the *M. smegmatis* genome for the existence of alternate enzymes that could perform this activity, as was found in *Escherichia coli*. However, in the context of target validation for drug discovery purposes, these results served to redirect our further efforts in experiments performed directly on *M. tuberculosis*, as detailed in the main text.

This explanation was adapted and introduced into the main text (lines 365-393 and 399-405).

10 - Ln 352-365: This information is not necessary in the discussion since this is well known and its removal makes the discussion more concise.

Thank you for the suggestion. We agreed that information is not necessary and it was removed.

11 - It would be useful to include a schematic of the shikimate pathway as a supplementary figure.

Thank you for the suggestion. The schematic of the shikimate pathway was added as a supplementary figure (Fig. S1).

12 - Figure 6 is a duplication of data (represented in a different way) and if considered necessary by the authors should be moved to supplementary.

The figure 6 was converted to Fig. S3.

Reviewer #2 (Comments for the Author):

The *aroG* gene encodes DAHP synthase of *M. tuberculosis*, the first step of the shikimate pathway which is essential for biosynthesis of aromatic amino acids, folates, mycobactins and menaquinol in mycobacteria. The shikimate pathway is known to be essential and as a result, DAHP synthase as well as other enzymes of the pathway have been investigated as drug targets. This work verifies that downregulation of *aroG* mRNA levels is inhibitory to *M. tuberculosis* in vitro as well as during growth in macrophages. Growth inhibition can be partially overcome by addition of aromatic amino acids with further addition of folate and mycobactin biosynthetic intermediates not further restoring growth but possibly even resulting in some growth inhibition for reasons that are not explored. The finding that *aroG* knockdown is inhibitory is not novel. Rescue of growth with aromatic amino acids is not surprising. Nevertheless, the work is well done.

We thank Reviewer #2 for the comments and suggestions that contributed to improve our work. In the response to comment 9 from Reviewer #1, we have detailed the reasons we think there is novelty in the data we provide.

Minor comments

1 - Fig. 3: include length of exposure (hours) in the legend (not mentioned in results section either).

Thank you for the suggestion, the legend and text were modified accordingly.

2 - Figure 4: inspection of the graphs does not really suggest that PAM1 is very different from PAM2 in terms of timing in inducing growth arrest.

Thank you for the suggestion. We ran the statistical analysis again and we confirmed that the significance in PAM1 appears in 72h with a P value = 0.04 whereas for PAM2 at 72h the P value = 0.09. For PAM2 there is statistical significance from 96h.

3 - The aromix supplement supplement consists of L-Trp, L-phe and L-tyr along with 4-hydroxybenzoic acid, 4-aminobenzoic acid and 2,3-dihydroxybenzoic acid. 4-aminobenzoic acid would be an intermediate in folate biosynthesis. 2,3-dihydroxybenzoic acid would be an intermediate in mycobactin biosynthesis but this would only be important under iron limitation. The 4-hydroxybenzoic acid is less clear since it's not an intermediate of menaquinol biosynthesis. Do we know that these do not inhibit an enzyme at 250uM? The concentration seems a bit high (combined it would be 750uM of benzoic acid derivatives added) although admittedly, rescue for Mtb is often only possible at high concentrations.

We can find in the literature reports of experiments to evaluate auxotrophy in strains of different bacteria (and also other organisms) deleted of different *aro* genes in which different compositions of “aromix” supplements were added. We recently shown that the growth impairment observed in *M. smegmatis aroA*-deficient cells could be completely rescued by solely adding the three aromatic amino acids (L-tryptophan, L-phenylalanine, and L-tyrosine), without additional supplementation with other aromatic compounds derived from chorismate (PMID: 34937164). In contrast, we show here that we were unable to completely rescue the growth impairment observed in *aroG*-deficient cells by supplementing them with only the three aromatic amino acids (Fig. 4). We obtained only a partial rescue, which prompted us to test whether the supplementation with additional aromatic compounds derived from chorismate would result in complete rescue. The aromix composition and concentrations were based on a previously published study (PMID: 12368440). Different aromix supplementations have been reported, without L-tyrosine (for example, PMID: 22892249, 24327559, 9066039, 12009288), 2,3-dihydroxybenzoate (PMID: 7927802) and, in most cases, without p-hydroxybenzoate (all the above references). We decided to be inclusive in our selection of aromix composition, to provide any chorismate derivative whose absence in the *aroG*-deficient strain could impart a growth defect. As observed by Reviewer #2, p-hydroxybenzoic acid is not an intermediate of menaquinol biosynthesis, and *M. tuberculosis* cells are deficient in the ubiquinones that are typically produced from p-hydroxybenzoic acid in other organisms. However, a chorismate pyruvate-lyase was isolated in *M. tuberculosis*, and the p-hydroxybenzoic acid was found to be used for the synthesis of glycoconjugates that are components of the bacterial cell wall and for the synthesis of phenolic glycolipids in some strains (PMID: 16210318).

To make this clear in our manuscript, we introduced the following piece of text in the Discussion section (lines 447-455):

“However, M. tuberculosis is thought to be deficient in ubiquinone, using only menaquinone in the electron transport chain. (Truglio et al, 2003). Despite that, p-hydroxybenzoate (a precursor of ubiquinones) is also produced from chorismate in M. tuberculosis, in a single-step reaction catalyzed by chorismate pyruvate-lyase (Stadthagen et al, 2005). In this organism, p-hydroxybenzoate seems to be used exclusively for the synthesis of p-hydroxybenzoic acid derivatives (p-HBADs), glycoconjugates that are components of the bacterial cell wall, and, in a limited number of strains, phenolic glycolipids (PGL) (Stadthagen et al, 2005).”

As noted above, the concentrations used for the aromatic compounds were the same reported previously (PMID: 12368440). After converting molarity to mg/ml, the sum of the three benzoic acid derivatives added is approx. 100 ug/ml, which is within the wide range of concentrations reported (from 10ug/ml - PMID: 7927802 – to 2 mg/ml - PMID: 22892249, 12009288).

June 24, 2022

Prof. Cristiano Valim Bizarro
Pontifícia Universidade Católica do Rio Grande do Sul
Centro de Pesquisas em Biologia Molecular e Funcional (CPBMF) and Instituto Nacional de Ciência e Tecnologia em Tuberculose (INCT-TB)
Porto Alegre, Rio Grande do Sul 90616-900
Brazil

Re: Spectrum00728-22R1 (Evaluation of 3-deoxy-D-arabino-heptulosonate-7-phosphate synthase (DAHPS) as a vulnerable target in *Mycobacterium tuberculosis*)

Dear Prof. Cristiano Valim Bizarro:

Your manuscript has been accepted, and I am forwarding it to the ASM Journals Department for publication. You will be notified when your proofs are ready to be viewed.

Sincerely,

Silvia Cardona
Editor, Microbiology Spectrum

Journals Department
Supplemental file 1: Accept